# Previous SARS-CoV-2 Infection Increases B.1.1.7 Cross-Neutralization by Vaccinated Individuals

**DOI:** 10.3390/v13061135

**Published:** 2021-06-12

**Authors:** Benjamin Trinité, Edwards Pradenas, Silvia Marfil, Carla Rovirosa, Víctor Urrea, Ferran Tarrés-Freixas, Raquel Ortiz, Jordi Rodon, Júlia Vergara-Alert, Joaquim Segalés, Victor Guallar, Rosalba Lepore, Nuria Izquierdo-Useros, Glòria Trujillo, Jaume Trapé, Carolina González-Fernández, Antonia Flor, Rafel Pérez-Vidal, Ruth Toledo, Anna Chamorro, Roger Paredes, Ignacio Blanco, Eulàlia Grau, Marta Massanella, Jorge Carrillo, Bonaventura Clotet, Julià Blanco

**Affiliations:** 1IrsiCaixa AIDS Research Institute, Germans Trias i Pujol Research Institute (IGTP), Can Ruti Campus, Autonomous University of Barcelona (UAB), 08916 Badalona, Spain; epradenas@irsicaixa.es (E.P.); smarfil@irsicaixa.es (S.M.); crovirosa@irsicaixa.es (C.R.); vurrea@irsicaixa.es (V.U.); ftarres@irsicaixa.es (F.T.-F.); rortiz@irsicaixa.es (R.O.); nizquierdo@irsicaixa.es (N.I.-U.); rparedes@irsicaixa.es (R.P.); egrau@irsicaixa.es (E.G.); mmassanella@irsicaixa.es (M.M.); jcarrillo@irsicaixa.es (J.C.); bclotet@irsicaixa.es (B.C.); 2Institute de Recerca i Tecnologia Agrària (IRTA), Centre de Recerca en Sanitat Animal (CReSA, IRTA-UAB), Campus de la UAB, 08193 Bellaterra, Spain; jordi.rodon@irta.cat (J.R.); julia.vergara@irta.cat (J.V.-A.); 3Centre de Recerca en Sanitat Animal (CReSA, IRTA-UAB), Campus de la UAB, Autonomous University of Barcelona (UAB), 08193 Bellaterra, Spain; joaquim.segales@irta.cat; 4Departament de Sanitat i Anatomia Animals, Facultat de Veterinària, UAB, 08193 Bellaterra, Spain; 5Barcelona Supercomputing Center, 08034 Barcelona, Spain; victor.guallar@bsc.es (V.G.); alba.lepore@bsc.es (R.L.); 6Catalan Institution for Research and Advanced Studies (ICREA), 08010 Barcelona, Spain; 7Fundació Althaia, Hospital de Sant Joan de Déu, 08243 Manresa, Spain; gtrujillo@althaia.cat (G.T.); jtrape@althaia.cat (J.T.); cgonzalezf@althaia.cat (C.G.-F.); aflor@althaia.cat (A.F.); rperez@althaia.cat (R.P.-V.); 8Infectious Diseases Department, Fight against AIDS Foundation (FLS), Germans Trias i Pujol Hospital, 08916 Badalona, Spain; rtoledo@flsida.org (R.T.); achamorro@flsida.org (A.C.); 9Germans Trias i Pujol Hospital, 08916 Badalona, Spain; iblanco.germanstrias@gencat.cat; 10Chair of Infectious Diseases and Immunity, University of Vic–Central University of Catalonia (UVic-UCC), 08500 Vic, Spain

**Keywords:** SARS-CoV-2, humoral response, pseudovirus, neutralization, B.1.1.7 variant

## Abstract

With the spread of new variants of severe acute respiratory syndrome coronavirus 2 (SARS-CoV-2), there is a need to assess the protection conferred by both previous infections and current vaccination. Here we tested the neutralizing activity of infected and/or vaccinated individuals against pseudoviruses expressing the spike of the original SARS-CoV-2 isolate Wuhan-Hu-1 (WH1), the D614G mutant and the B.1.1.7 variant. Our data show that parameters of natural infection (time from infection and nature of the infecting variant) determined cross-neutralization. Uninfected vaccinees showed a small reduction in neutralization against the B.1.1.7 variant compared to both the WH1 strain and the D614G mutant. Interestingly, upon vaccination, previously infected individuals developed more robust neutralizing responses against B.1.1.7, suggesting that vaccines can boost the neutralization breadth conferred by natural infection.

## 1. Introduction

Early in the COVID-19 pandemic, severe acute respiratory syndrome coronavirus 2 (SARS-CoV-2) variants started to develop regionally and globally. Currently, the rapid spread of the B.1.1.7, or 501Y.V1, variant [1], first reported in the UK, casts doubts on the protection conferred by the neutralizing antibody response acquired during a previous immunization. Besides the D614G mutation, the B.1.1.7 variant contains six non-synonymous mutations and three deleted amino acids in the spike (S) protein (Figure 1A). The major changes are the mutation N501Y in the receptor-binding domain (RBD); the deletion 69–70 which may increase transmissibility [2] and produces a false negative in certain RT-PCR-based diagnostic assays; and the mutation P681H, next to the furin cleavage site, that could impact antigenicity and enhance viral infectivity. Recent studies indicate that B.1.1.7 is associated with a higher hospitalization risk [3] and higher mortality [4] and several reports indicate that it has a higher secondary attack rate, making this viral variant 30–50% more transmissible [5]. Importantly, this variant remains susceptible to some monoclonal and plasma antibodies from convalescent or vaccinated individuals [6,7,8].

Here we analyzed cross-neutralizing plasma antibody titers in individuals infected during both the first and second waves of COVID-19 epidemics in Catalonia (Spain), as well as in vaccinated individuals. The B.1.1.7 variant showed minimal resistance to the neutralizing capacity from both infected and vaccinated individuals, but its impact was significantly more pronounced on the latter group. Interestingly, previous infection significantly improved neutralization titers against this variant upon vaccination.

## 2. Materials and Methods

### 2.1. Study Overview and Subjects

The study was approved by the Hospital Ethics Committee Board from Hospital Universitari Germans Trias i Pujol (PI-20-122 and PI-20-217) and all participants provided written informed consent before inclusion.

Plasma samples were obtained from the prospective KING cohort of the HUGTiP (Badalona, Spain) and from Althaia (Manresa, Spain). The KING cohort included individuals with a documented positive RT-qPCR result from nasopharyngeal swab and/or a positive serological diagnostic test.

Samples in this study were collected from March 2020 to February 2021; thus, covering the different COVID-19 outbreaks in Catalonia (dadescovid.cat). We analyzed 32 non-vaccinated individuals infected in March 2020, using plasma samples collected at a median of 48 days (*n* = 16) or 196 (*n* = 16) days after symptom onset. We also selected 16 individuals infected in August 2020 using plasma samples collected at a median of 44 days after symptom onset, and 5 patients (*n* = 13 samples) infected in January 2021 by the B.1.1.7 variant. Finally, 32 individuals having received two doses of Pfizer/BioNTech vaccine were sampled 2 weeks after the second dose. This last group included uninfected and long-term previously-infected individuals. A description of the different groups and subgroups is shown in Table 1.

### 2.2. Cell Lines

HEK293T cells (presumably of female origin) overexpressing WT human ACE-2 (Integral Molecular, Philadelphia, PA, USA) were used as a target for a SARS-CoV-2 spike-expressing pseudovirus infection. The cells were maintained in T75 flasks with Dulbecco′s Modified Eagle′s Medium (DMEM) supplemented with 10% FBS and 1 µg/mL of Puromycin.

### 2.3. Spike Plasmid Generation

SARS-CoV-2.SctΔ19 WH1 and B.1.1.7 were generated (Geneart) from the full protein sequence of the original SARS-Cov-2 isolate Wuhan-Hu-1 (WH1) and the UK variant (B.1.1.7) spike sequences respectively, with the deletion of the last 19 amino acids in C-terminal [9], human-codon optimized and inserted into pcDNA3.1(+). The D614G spike mutant was generated by site-directed mutagenesis as previously described [10]. In brief, SARS-COV-2.SctΔ19 WH1 plasmid was amplified by PCR with Phusion DNA polymerase (Thermo Fisher Scientific, Waltham, MA, USA, ref# F-549S) and the following primers: 5′-TACCAGGgCGTGAACTGTACCGAAGTGCC-3′ and 5′-GTTCACGcCCTGGTACAGCACTGCCAC-3′. PCR was 20 cycles with an annealing temperature of 60 °C and an elongation temperature of 72 °C. The PCR product was then treated for 3 h with the DpnI restriction enzyme (Thermo Fisher Scientific, ref# ER1705), to eliminate template DNA, and transformed into supercompetent *E. coli*. The final mutated DNA was then fully sequenced for validation.

### 2.4. Pseudovirus Generation and Neutralization Assay

HIV reporter pseudoviruses expressing SARS-CoV-2 S protein and Luciferase were generated using the defective HIV plasmid pNL4-3.Luc.R-.E- obtained from the NIH AIDS Reagent Program [11]. Expi293F cells were transfected using ExpiFectamine293 Reagent (Thermo Fisher Scientific) with pNL4-3.Luc.R-.E- and SARS-CoV-2.SctΔ19 (WH1, G614 or B.1.1.7), at an 8:1 ratio, respectively. Control pseudoviruses were obtained by replacing the S protein expression plasmid with a VSV-G protein expression plasmid as reported [12]. Supernatants were harvested 48 h after transfection, filtered at 0.45 µm, frozen and titrated on HEK293T cells overexpressing WT human ACE-2. Spike expression in pseudovirus-producing cells was confirmed by flow cytometry, showing comparable expression for all constructs (Figure 1B).

Neutralization assays were performed in duplicate. Briefly, in Nunc 96-well cell culture plates (Thermo Fisher Scientific), 200 TCID_50_ of pseudovirus were preincubated with three-fold serial dilutions (1/60–1/14,580) of heat-inactivated plasma samples for 1 h at 37 °C. Then, 2 × 10^4^ HEK293T/hACE2 cells treated with DEAE-Dextran (Sigma-Aldrich) were added. Results were read after 48 h using the EnSight Multimode Plate Reader and BriteLite Plus Luciferase reagent (Perkin Elmer, Waltham, MA, USA). The values were normalized, and the ID_50_ (the reciprocal dilution inhibiting 50% of the infection) was calculated by plotting and fitting the log of plasma dilution vs. response to a 4-parameter equation in Prism 8.4.3 (GraphPad Software, San Diego, CA, USA). This neutralization assay had been previously validated in a large subset of samples [13,14]. The lower limit of detection was 60 and the upper limit was 14,580 (reciprocal dilution).

### 2.5. Flow Cytometry

Transfected Expi293F cells were first stained extracellularly with a polyclonal rabbit anti-spike RBD antibody (Sino Biological, Beijing, China, ref# 40592-T62) and a secondary APC labeled anti-rabbit (Jackson Immuno Research, West Grove, PA, USA, ref# 711-605-152). Cells were then fixed (Life Technologies, Carlsbad, CA, USA, ref# GAS001S100) and stained in permeabilization buffer (Life Technologies, ref# GAS002S100), with a FITC-labeled mouse anti-p24Gag antibody KC57 (Beckman Coulter, Brea, CA, USA, ref# 6604665). Cells were acquired on a BD Celesta flow cytometer with DIVA software and analyzed on FlowJo vX.0.7 (Tree Star, Inc., Ashland, OR, USA).

### 2.6. Statistical Analysis

Continuous variables were described using medians and the interquartile range (IQR, defined by the 25th and 75th percentiles), whereas categorical factors were reported as percentages over available data. Quantitative variables were compared using the Mann–Whitney test; and percentages using the chi-squared test. The Friedman test with Dunn’s multiple comparison test was used to compare neutralization of different pseudoviruses. Multiple M–W comparisons were corrected by false discovery rate. Analyses were performed with Prism 8.4.3 (GraphPad Software) and R version 4.0 (R Foundation for Statistical Computing).

## 3. Results

### 3.1. Global Analysis of Cross-Neutralizing Titers in Vaccinated and Infected Participants

As a first approach, we tested the neutralizing antibody response of all participants segregated in two main groups labeled “all vaccinated” and “all infected”. “All vaccinated” included all the vaccinated participants whether they had experienced previous natural infection or not. “All infected” included all non-vaccinated participants who were infected during either the first wave, the second wave or specifically infected by the B.1.1.7 variant. We tested all the plasma samples (*n* = 98) against pseudoviruses expressing three different spike glycoproteins: a spike corresponding to the original SARS-CoV-2 virus, isolated in Wuhan, and named here WH1; a D614G mutant based on the WH1 spike and a spike including the defining mutations of the B.1.1.7 variant and named B.1.1.7.

The neutralization titers of all vaccinated individuals against B.1.1.7 and WH1 were not statistically different (according to the Friedman test with Dunn’s multiple comparison test) (Figure 2A), despite WH1 expressing the spike sequence on which the vaccine was based. In contrast, vaccinated individuals showed significantly higher potency to neutralize the intermediate D614G mutant (*p* < 0.0001) (Figure 2A). A similar analysis, including all non-vaccinated infected individuals, showed similar results (Figure 2B). The highest neutralization (*p* < 0.0001) was noticed for the D614G mutant, while no significant differences were observed between WH1 and B.1.1.7. Next, we compared cross-neutralization capacities by determining fold change ratios between a spike of interest (S^x^, indicated on top) and a reference spike (S^r^, indicated in the Y axis of Figure 2C,D, fold change = S^r^/S^x^). This ratio is a measure of the loss of neutralizing capacity; a ratio inferior to one indicates a better neutralization of the spike of interest in comparison to the reference spike, and vice versa. When B.1.1.7 and WH1 variants were compared, median fold-change was 0.97 in the vaccinated group, a value significantly higher than the one obtained in naturally infected individuals (0.7; *p* = 0.033, M–W test, Figure 2C). Similarly, the fold change between B.1.1.7 and D614G was significantly different between the all vaccinated and the all infected groups (median values 1.9 vs 1.17, respectively; *p* < 0.0001, M–W test, Figure 2D). Altogether, these data indicate that, relative to WH1 and D614G, cross-neutralization of B.1.1.7 was worse in vaccinated individuals in comparison to infected ones.

### 3.2. Identification of Parameters Influencing Cross-Neutralization

To better understand these differences, we analyzed infected and vaccinated subgroups. To assess the impact of sequence evolution on the immune responses, infected individuals were divided according to infection date. Individuals infected during the first wave (March 2020) in Spain were initially exposed to the original D614 virus that was rapidly displaced by the G614 variant. An evolving 85 to 22% prevalence for the original variant has been estimated [15]. Individuals infected during the second wave (August 2020) were almost exclusively exposed to the G614-containing 20E (EU1) lineage, which accounted for nearly 100% of new infections during the summer of 2020 (https://nextstrain.org/ncov/global, accessed on 31 March 2021). Individuals infected by the B.1.1.7 variant, identified in January 2021, were also analyzed.

First, we analyzed the neutralization response specifically in participants infected during the first wave. Individuals sampled 48 days after infection showed a small but significant decrease in neutralization capacity against the B.1.1.7 variant when compared with the D614G mutant (median fold change of 1.53, *p* = 0.031, Friedman test, Figure 3A,B). When compared to WH1, no significant difference was observed (median = 1.20-fold, *p* = 0.155). Individuals infected on March 2020 and sampled 6 months later also showed a general decay of the neutralization response as previously reported by us and many others [14,16]. Interestingly though, in these limited cohorts, decay of the neutralization response was only significant when measured against WH1 (*p* = 0.0374, K-W, Figure 3A) and not against D614G or B.1.1.7. In fact, we observed a general trend of improving cross-neutralization capacities against both D614G and the B.1.1.7 variant when compared with WH1, although it was only significant for the latter (fold change evolving from 1.2 to 0.6, *p* = 0.024, M–W, Figure 3B). These data suggest that the neutralization response induced by natural infection continued to evolve several months after the initial symptoms.

We then looked at the neutralization responses induced during the subsequent waves of infection. Individuals infected in August 2020 or infected by the B.1.1.7 variant behaved differently from those infected in March 2020, in that no significant differences were detected between neutralizing titers against all three tested spikes (Figure 3C). When comparing ratios with first wave participants, B.1.1.7 infected individuals showed a significant increase in the neutralization of B.1.1.7 vs D614G (fold change going from 1.53 to 1, *p* = 0.0006) (Figure 3D). These data illustrate the progressive evolution of the virus infecting the population and its impact on the induced cross-neutralization response. Importantly, individuals infected by the B.1.1.7 variant were still able to cross-neutralize the original WH1 spike and the D614G mutant.

Finally, we analyzed vaccinated individuals. Vaccinated individuals were sub-classified according to previous COVID-19 evidence into infected or uninfected. We compared three groups: vaccinated non-infected, vaccinated previously infected during the first wave and the group of first wave infected participants (late plasma sampling) not vaccinated. As we did not have the neutralization levels of vaccinated previously infected individuals prior to vaccination, we used the group of first wave infected (late plasma) individuals as a surrogate.

When analyzing specifically vaccinated individuals not previously infected, the decrease in B.1.1.7 cross-neutralization already observed for the all vaccinated group (Figure 2A) was even more clear (Figure 4A). This was significant in comparison to both WH1 (fold change of 2.04, *p* = 0.0021) and D614G (fold change of 2.65, *p* = <0.0001). Vaccinated individuals previously infected did not show such a decrease; in fact, B.1.1.7 neutralization was increased compared to WH1 (fold change of 0.60, *p* = 0.0034). B.1.1.7 neutralization was still reduced when compared to D614G (fold change of 1.55, *p* = 0.0027) but, overall, the response in the vaccinated previously infected group was improved in comparison to vaccinated only individuals (Figure 4A,B).

Neutralization titers in infected only individuals were generally lower than in the vaccinated groups (Figure 4A) as the former group experienced a decay in the neutralizing response over time (Figure 3A) [14,16] while vaccinated participants were sampled between 1 and 2 weeks after the second shot, when the neutralizing response was maximized [17]. However, in comparison to vaccinated only individuals, the cross-neutralizing response against both D614G and B.1.1.7 was superior to WH1 and the response against B.1.1.7 was only marginally reduced compared to D614G (fold change of 1.28, *p* = 0.034). Interestingly, the cross-neutralization profile of vaccinated previously infected individuals was very similar to infected only participants as there was no significant difference between the two groups when looking at the ratios of the neutralizing response between variants (Figure 4B). This suggests that vaccination was able to boost the full cross-neutralization breadth of the response previously established by natural infection during the first wave. In fact, in these limited cohorts, the averages of the titers against each variant were all increased by a factor of four when considering the average titers of vaccinated and non-vaccinated previously infected groups (Figure 4A).

## 4. Discussion

D614G neutralization was significantly increased in both vaccinated and infected, compared to WH1. This is in agreement with recent reports showing that D614G mutation is associated with a more open (one-up) conformation which increases access to the RBD and results in both an increase in infectivity [18] and an increase in sensitivity to neutralization [19,20]. Interestingly, this effect was increased when testing the plasma at 6 months from infected individuals but reduced when testing plasma from the second wave (largely G614) as well as B.1.1.7 infected participants.

Overall, we show that the B.1.1.7 variant minimally impacted sensitivity to neutralizing immune responses from individuals infected during the first wave and sampled between 1 and 2 months after symptoms, in line with other reports [6,21]. The neutralizing response associated with the B.1.1.7 variant only showed a significant decrease when compared to the D614G mutant but not with the original WH1 spike. Importantly, these small differences faded in first wave participants tested 6 months after infection, despite a general decay in the neutralization response. This observation supports the positive evolution over time of neutralizing responses in infected individuals suggested by different authors [14,22].

No significant impact of the B.1.1.7 variant was observed when analyzing the neutralizing activity of participants infected during the second wave, in August 2020. During the second wave in Spain, G614 genotype was highly prevalent but the B.1.1.7 variant was still absent. This suggests that the intermediate evolution of the virus spike sequence was sufficient to increase the quality of the neutralizing response against the B.1.1.7 variant. The second wave participants’ plasma was collected between 1 and 2 months following symptoms. Considering our observations on the evolution of the cross-neutralization capacities in first wave participants, it will be interesting to perform a similar follow up on second wave participants. As expected, participants specifically infected by the B.1.1.7 variant, in January 2021, demonstrated the best cross-neutralization capacity against the B.1.1.7 variant in comparison to other spikes. Their plasma was collected earlier than the other groups, between 1 and 3 weeks following symptom onset. Once again, additional follow ups will refine our data. Finally, in line with other reports [23], we demonstrate that infection by the G614-containing 20E (EU1) and B.1.1.7 variants elicits cross-neutralizing responses against former viral variants.

Finally, we show here that vaccinated individuals, as a whole, suffered a small reduction of cross-neutralizing activity against B.1.1.7 in comparison to D614G, as previously reported [24,25,26]. This loss of neutralization was more evident when excluding vaccinated individuals who had experienced a previous SARS-CoV-2 infection. Indeed, previously infected vaccinated participants demonstrated a much-improved neutralization of the B.1.1.7 variant. The capacity of this group to neutralize B.1.1.7 in comparison to both WH1 and D614G was not statistically different from infected individuals tested after 6 months. This suggests that, even though the vaccine alone did not generate an optimal B.1.1.7 neutralization response (compared to natural infection), it was capable of boosting the full cross-neutralization response pre-established by natural infection, in line with other reports [27,28]. Importantly, vaccinated individuals were all sampled at least 7 days after the second dose of vaccine, at a time when both humoral and neutralizing responses have reportedly reached a plateau [29,30]. However, another study suggests that the kinetics of neutralization following vaccination might vary depending on the variant [31]. Moreover, our data do not allow us to draw any conclusion on the long-term evolution of immune responses elicited by vaccines, in terms of the quality or durability of antibodies.

In conclusion, the neutralizing response from infected individuals, while slowly decaying in magnitude, seems to show a good qualitative evolution which can be fully recalled upon vaccination. Importantly, our data suggest a better cross-neutralizing quality of antibodies induced by natural infection compared to those induced exclusively by vaccination. It will be interesting to analyze how neutralization breadth evolves over time in non-infected vaccinated individuals to define if new vaccines will be necessary to further enrich it.

## Figures and Tables

**Figure 1 viruses-13-01135-f001:**
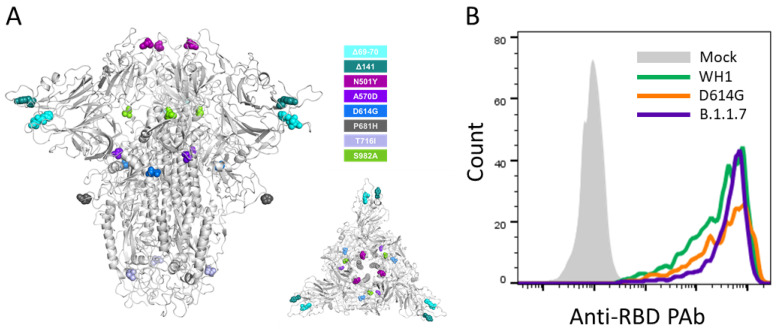
In vitro infectivity of SARS-CoV-2 variants. (**A**). The different mutations identified in the B.1.1.7 variant are listed and their location in the spike protein (side and top views) is shown. This variant also includes the D614G mutation. (**B**). Spike expression in pseudovirus-producing cells stained with an anti-RBD polyclonal rabbit antibody (See Methods for details).

**Figure 2 viruses-13-01135-f002:**
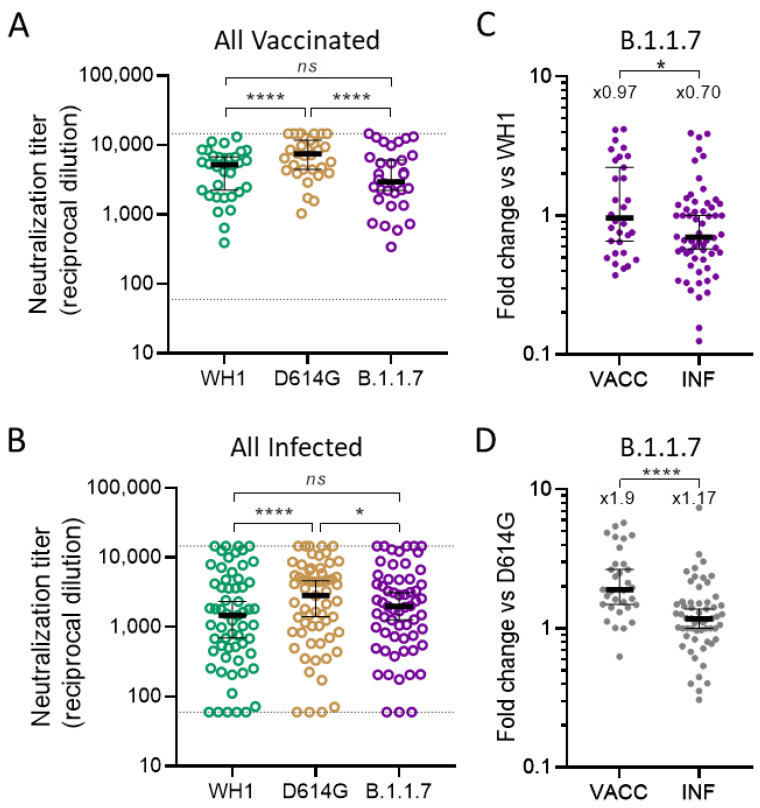
Global analysis of neutralization titers in SARS-CoV-2 vaccinated and infected individuals. Values of ID_50_ (as reciprocal dilution) are shown for all plasma samples from (**A**) vaccinated and (**B**) infected non-vaccinated individuals against the indicated pseudoviruses. Bars indicate median titer in each group with a 95% confidence interval and *p* values show the comparison of median titers among the three viruses (Friedman test with Dunn’s multiple comparison test, * *p* < 0.05, **** *p* < 0.0001). The corresponding fold-change in neutralization titers between (**C**) WH1 and B.1.1.7 or (**D**) D614G and B.1.1.7 is shown (lower is better), comparing the vaccinated (VACC) and the infected (INF) groups. Bars indicate median in each group with 95% confidence interval and top values indicate the median fold-change between the indicated variants (variants compared are indicated in the graph title and in the Y axis). Fold change medians were compared using the Mann–Whitney test (* *p* < 0.05, **** *p* < 0.0001).

**Figure 3 viruses-13-01135-f003:**
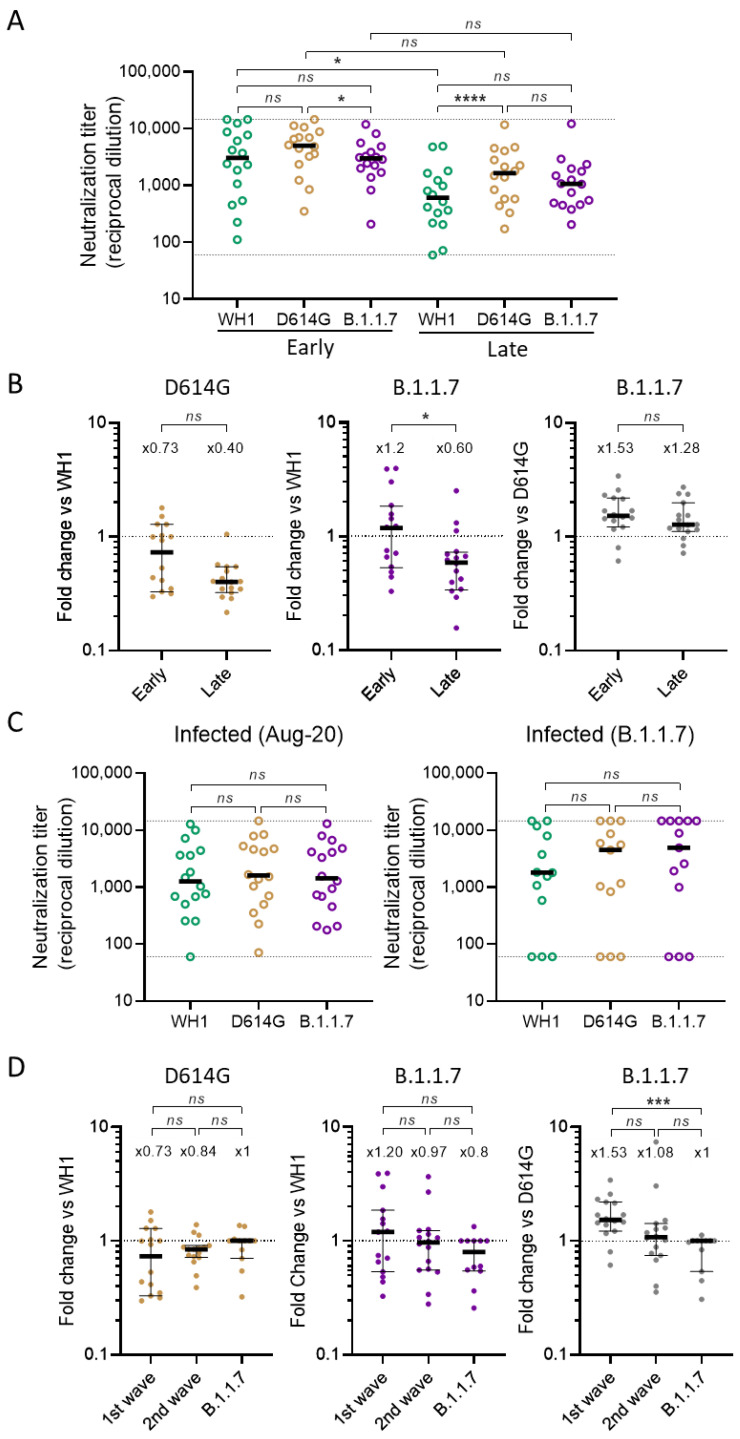
Subgroup analysis of neutralization titers in infected individuals. (**A**,**B**). Comparison of plasmas collected at 48 days (early sampling) and >6 month (late sampling) after infection from 2 independent groups of participants infected in March 2020 (first wave). Neutralization titers (ID_50_ expressed as reciprocal dilutions) are shown in (**A**), and the corresponding ratios between variants (lower is better) are shown in (**B**). (**C**,**D**) Neutralizing titer of individuals infected during the second wave (August 2020) or specifically by the B.1.1.7 variant. Neutralization titers (ID_50_ expressed as reciprocal dilutions) are shown in (**C**), and the corresponding ratios between variants (lower is better) are shown in (**D**). In (**D**), we also included ratios from the first wave infected individuals (early sampling) for comparison. In (**A**,**C**), bars indicate median titer in each group and *p* values show the comparison of median titers among the three viruses (Friedman test with Dunn’s multiple comparison test; * *p* < 0.05, **** *p* < 0.0001). Specifically in (**A**), *p* values are also indicated for the comparison of neutralization titers against the same spike between the 2 groups (Kruskal–Wallis test; * *p* < 0.05). In (**B**,**D**), bars indicate the median in each group with a 95% confidence interval and top values indicate the median fold-change between the indicated variants (variants compared are indicated in the graph title and in the Y axis). *p* values show the comparison of the group sampled at 48 days vs the group sampled at 6 months (Mann–Whitney test; * *p* < 0.05). In (**D**), *p* values show the comparison between individuals from the different infection waves (Kruskal–Wallis with Dunn’s multiple comparison test; *** *p* < 0.001).

**Figure 4 viruses-13-01135-f004:**
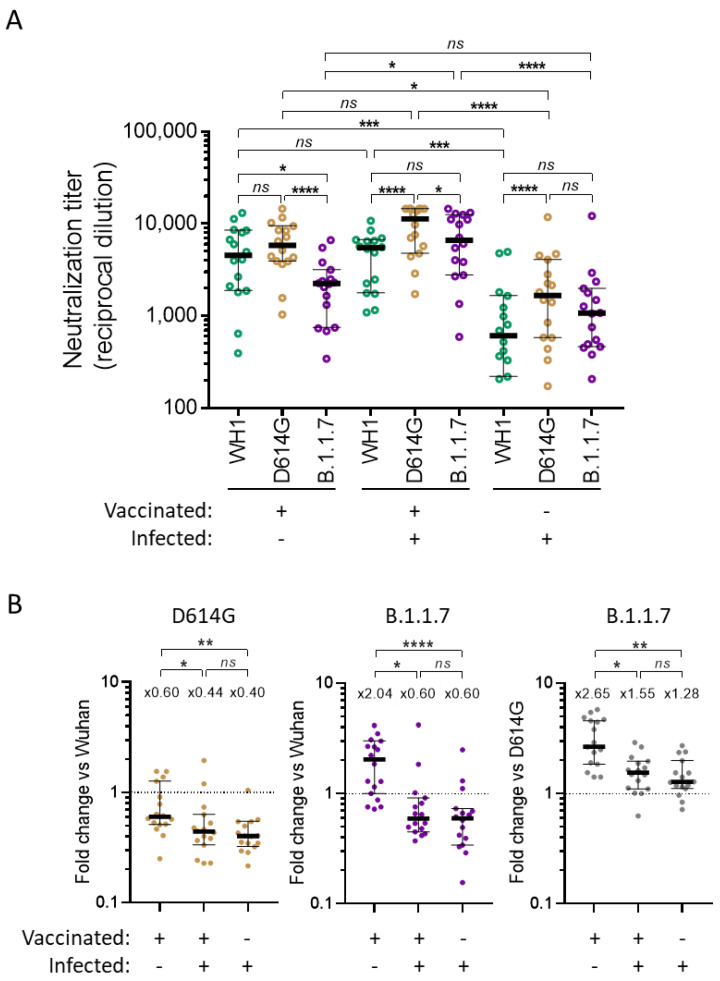
Subgroup analysis of neutralization titers in vaccinated individuals. Comparison of plasma from vaccinated individuals, previously infected or not during the first wave, as well as late plasma from first wave infected individuals. (**A**) Neutralization titers (ID_50_ expressed as reciprocal dilutions). Bars indicate the median titer in each group with a 95% confidence interval. *p* values show the comparison of median titers against the three variants in the same group (Friedman with Dunn’s multiple comparison test ; * *p* < 0.05, **** *p*  <  0.0001) and the comparison of the response against the same spike between groups (Kruskal–Wallis with Dunn’s multiple comparison test * *p* < 0.05, *** *p* < 0.001, **** *p*  <  0.0001) (**B**) Corresponding ratios between variants (lower is better). Bars indicate the median in each group with a 95% confidence interval and top values indicate the median fold-change between the indicated variants (variants compared are indicated in the graph title and in the Y axis). *p* values show the comparison of median ratios between each group (Kruskal–Wallis with Dunn’s multiple comparison tests; * *p* < 0.05, ** *p* < 0.01, **** *p* <  0.0001).

**Table 1 viruses-13-01135-t001:** Description of participants. Uninfected individuals were included as negative controls for neutralizing activity. All of them showed undetectable neutralizing activity. IQR: interquartile range.

	Uninfected	Infected Non-Vaccinated				Vaccinated	
	*n* = 5	*n* = 53				*n* = 32	
Infection Status	Uninfected	Infected		Infected	Infected	Infected	Uninfected
Date of Infection		March 2020		August 2020	January 2021	March 2020	
Strains		D614/G614		20E (EU1)	B.1.1.7	D614/G614	
Sampling		Early	Late				
		*n* = 16	*n* = 16	*n* = 16	*n* = 5	*n* = 16	*n* = 16
Age (years), median (IQR)	46 (42–52)	65 (55–68)	56 (54–62)	44 (37–54)	79 (60–91)	39 (29–44)	45 (30–61)
Gender (female), *n* (%)	4 (80)	4 (25)	7 (44)	8 (50)	2 (40)	11 (69)	12 (75)
Days from symptom onset, median (IQR)	---	48 (36–57)	196 (186–207)	44 (37–54)	16 (8–20) *	324 (184–339)	---
Days from vaccination, median (IQR)	---	---	---	---	---	13 (10–14)	9 (7–12)
Hospitalized, *n* (%) Severe (%)	--- ---	11 (69) 6 (38)	11 (69) 6 (38)	11 (69) 5 (31)	5 (100) 0 (0)	0 (0) 0 (0)	--- ---

* Data from 13 samples obtained from 5 participants in this group.

## Data Availability

Data are not publicly posted. They can be shared upon request to corresponding authors.

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
