# Peer review of "Previous SARS-CoV-2 Infection Increases B.1.1.7 Cross-Neutralization by Vaccinated Individuals"

_viruses, 2021, doi:10.3390/v13061135_

Round 1

Reviewer 1 Report

The present report by Trinite et al., is an exciting piece of work adding information about neutralization antibodies developed during past waves of infections in Spain plus the effect of vaccination on the newer surfacing strains. This study will be an important piece of information to the community. With high level of enthusiasm there are serious caveats in the presentation and interpretation of the results which authors need to work upon as they must realize the community needs to understand this piece of work. 

Major Points:

  1. The paper talked about neutralization of the virus all through but what actually is neutralizing them was not mentioned- for example the sequence of IgG specific for the viruses etc. The authors need to be more descriptive in their approach to make readers interpret their results.
  2. Neutralization assays only inform about antibodies developed but what about the T-cells and B-cell memory or other NK-cells? Did the authors have access to such resources to make a wholistic conclusion apart from just antibodies?
  3. SARS-Cov2 has a partial bias based on sex and production of antibodies like in case of influenza. More female donors were considered for vaccinated group, was it an intentional strategy? It would be good if the authors can explain their findings based on this bias- how the male and female population responding to each.

Minor Comments but very important and authors must take care of this part:

  1. Fig. 2B mention is missing. When you are presenting a data you need to mention a few lines, just cant afford to skip.
  2. The result section needs to have a short description of the interpretation of the analysis. The readers wont wait for understanding the results from discussion section. 
  3. When labeling in figure legends: it should be p-value and not P-value. 
  4. All figure description in results have been very loosely described. Missing most of the interpretations. Authors must describe each and every graph spending atleast 4-5 lines. 
  5. For example, authors say Line 205-206: "showed a general trend of improving neutralization titers against B.1.1.7 variant-" and then tags this statement with Fig. 3A. Clearly nothing of such sort is observable in 3A but in 3B. This needs professional handling. 

Reviewer 2 Report

Summary

In the study by Trinité et al., the group tested neutralizing activity of the plasma from infected and/or vaccinated individuals in Catalonia (Spain) against pseudoviruses expressing the Spike of the original Wuhan strain, the D614G mutant and the B.1.1.7 variant.

They showed that the B.1.1.7 variant had minimal resistance to the neutralizing capacity of infected and vaccinated individuals, which is significantly more pronounced on the latter group. In addition, they observed that the previous infection significantly improved neutralization titers against this variant upon vaccination.

Overall, this manuscript offers interesting data on the COVID-19 epidemics in Catalonia (Spain) and is well written. However, there are a few comments that need to be investigated before publication.

Major comments

Materials and Methods

  1. The description of participants in the Table 1 is not so clear. Authors should add some missing information to help the reader as the strain of infection following different time collection, or a clear discrimination of already infected participants in each group will be appreciated.
  2. What is the detection limit of your neutralization assays? Authors should add this information in the Materials and Methods

Results

  1. Regarding the participants of your study, what type of severity of SARS-CoV-2 infection did they develop? Are there any differences noted in the neutralizing capacity of people with asymptomatic severe infection? How did you discriminate the different levels of infection (asymptomatic, severe, critical)?
  2. Plasma from uninfected vaccinated individuals were collected earlier than from infected vaccinated donors, does timing have an impact on the capacity to neutralize?
  3. Why represent neutralization titer by geometric mean? Median should be better.
  4. Figure 2, why not perform the same fold change analysis or the two groups?
  5. General comment on the figures, authors should represent linked dot plots (symbols and lines) per donor to see information per donor.
  6. General comment on the legends, authors should mention the interpretation of fold change in the text and not in the legend “lower is better”.
  7. Figure 3, can you represent, on a same graph, the comparison between the different strains early late?
  8. Figure 4A, a lot of inter and intra-group comparisons are missing.

Discussion

  1. Do you have information on Antibodies quantification and ADCC function on these donors? If yes, do you see correlations with neutralization functions as it is already published?
  2. Authors should include in their conclusions the timing of plasma collection which might play a role in the detection of the neutralization capacity (as it is already published by other groups).
  3. Do you have other time points to analyze between the two doses of vaccine?
  4. Line 297, authors used “small reduction” but information are missing to confirm this as this is not a longitudinal analysis, authors should temper this sentence.
  5. Some recent important publications are missing from the discussion, in particular the paper of Planas et al. (PMID: 33772244) or the work of Andres Finzi lab (PMID: 33969322 ;PMID: 33758857).

Minor comments

  1. Line 172, remove one “between”.

Reviewer 3 Report

This work by Trinite and Pradenas et al. has investigated the cross-neutralization activities of the produced antibodies from the infected, vaccinated, or both infected and vaccinated individuals.

COVID-19 affects to all over the world, and it is an urgent matter to prevent the spread. There is no doubt that vaccination could produce neutralization antibody to protect from the individual infection.

In the current study, authors survey the protective capacity of the antibody produced from infected, vaccinated, or both infected and vaccinated individuals. Authors examined their neutralization activities against original Wuhan strain, the D614 mutant, and B.1.1.7 variant using pseudotype virus.

Overall, the study was well organized and showed that upon vaccination, previously infected individuals developed more robust neutralization antibody against B.1.1.7. The results supported author’s conclusion. However, it is not clear what is novel compared to the previous studies, which is critical to publish in an original research article. Authors should emphasize the novelty, the difference compared to the other studies.

There are some minor points to be edited.

・”p=<0.0001” should be” p<0.0001” (ex. Page 4, line 164, page 5 line 166 and 168).

・”between between” should be “between” (page 5, line 173).

・Page 5, line 173-175, it might be better to describe “Altogether, this data indicated that the degree of the neutralization of B.1.1.7 was modestly lower on Vaccinated individuals in comparison to infected ones.”

・”However, In comparison” should be “However, in comparison” (page 10, line 267).

Overall, the manuscript was well designed and written, but few more information would assist readers to understand the advantage of the study.

Reviewer 4 Report

The manuscript by Trinité et al. analyzed neutralization titer of plasma from convalescent patients and vaccinees using pseudovirus viruses containing the Wuhan reference, D614G, or B.1.1.7 spike in the virion surface. Neutralization data showed broad neutralization against these pseudoviruses by either infected or vaccinated sera. In general, D614G variant is more susceptible to both vaccinated and previously infected sera while B.1.1.7 has similar sensitivity to these sera as the wildtype strain. The authors also showed a slight decay on neutralizing potency between early sampling and late sampling sera. However, the neutralization activity against B.1.1.7 was less decreased comparing to that against wildtype strain. Interestingly, the authors showed a previous infection plus a vaccination increased the potency on neutralizing these pseudoviruses comparing to vaccination only or infection only. the manuscript is well written and the data reported here are welcomed complementary and consistent to recent reports cited in this study. However, there are several concerns should be addressed before further consideration.

  1. 5 line 173-174, the data in Figure 2C and 2D do not support the conclusion made here. The larger fold change as seen for infected sera indicates more enhanced potency against B.1.1.7 comparing wildtype reference strain. This does not necessarily suggest vaccinated sera has reduced neutralization against B.1.1.7 since vaccinated sera are more potent than convalescent sera on neutralizing wildtype pseudovirus.
  2. A direct comparison of neutralization potency (rather than fold change) between early, late, and B.1.1.7 infection as shown in Figure 3 would be more beneficial for the audience to understand whether there is difference in immune response to these infections.
  3. While the protection against B.1.1.7 is less decayed comparing to wild type between early sampling and late sampling as shown in Figure 3, investigation on other variants of concern such as B.1.351 is much appreciated.
  4. It is not clear how the fold change been defined. Maybe more elaboration on this could make the concept more clear to the audience.
  5. The strain name should be used rather than city name for a virus according to WHO best nomenclature practice.
  6. Figure 3D, middle panel, the fold change in 1st wave appears to be >1 but labelled as “x0.20”.
  7. Line 62, the sequence used in mRNA vaccine is not the same to the wildtype reference sequences. So It may be better to revise the wording here to avoid confusion.
  8. Line 208, Figure 3B rather Figure 3A than should be recalled here.

Round 2

Reviewer 2 Report

I thank the authors for the remarkable work they have done in their response. All my concerns have been taken into account.

Reviewer 3 Report

I do not have further comment.